# Provably Efficient Causal Model-Based Reinforcement Learning for Systematic Generalization

## Abstract

In the sequential decision making setting, an agent aims to achieve *systematic generalization* over a large, possibly infinite, set of environments. Such environments are modeled as discrete Markov decision processes with both states and actions represented through a feature vector. The underlying structure of the environments allows the transition dynamics to be factored into two components: one that is environment-specific and another one that is shared. Consider a set of environments that share the laws of motion as an illustrative example. In this setting, the agent can take a finite amount of *reward-free* interactions from a subset of these environments. The agent then must be able to *approximately* solve any planning task defined over any environment in the original set, relying on the above interactions only. Can we design a provably efficient algorithm that achieves this ambitious goal of systematic generalization? In this paper, we give a partially positive answer to this question. First, we provide the first tractable formulation of systematic generalization by employing a *causal* viewpoint. Then, under specific structural assumptions, we provide a simple learning algorithm that allows us to guarantee any desired planning error up to an unavoidable sub-optimality term, while showcasing a polynomial sample complexity.

## 1 Introduction

Whereas recent breakthroughs have established Reinforcement Learning (RL) Sutton & Barto (2018) as a powerful tool to address a wide range of sequential decision making problems, the curse of generalization Kirk et al. (2021) is still a main limitation of commonly used techniques. RL algorithms deployed on a given task are usually effective in discovering the correlation between an agent's behavior and the resulting performance from large amounts of labeled samples. However, those algorithms are usually unable to discover basic cause-effect relations between the agent's behavior and the environment dynamics. Crucially, the aforementioned correlations are oftentimes specific to the task, and they are unlikely to be of any use for addressing different tasks. Instead, some universal causal relations generalize over the environments, and once learned can be exploited for solving any task. Let us consider as an illustrative example an agent interacting with a large set of physical environments. While each of these environments can have its specific dynamics, we expect the basic laws of motion to hold across the environments, as they encode general causal relations. Once they are learned, there is no need to discover them again from scratch when facing a new task, or an unseen environment. Even if the dynamics over these relations can change, such as moving underwater is different than moving in the air, or the gravity can change from planet to planet, the underlying causal structure still holds. This knowledge alone often allows the agent to solve new tasks in unseen environments by taking a few, or even zero, interactions.

We argue that we should pursue this kind of generalization in RL, which we call *systematic generalization*, where learning universal causal relations from interactions with a few environments allows us to approximately solve any task in any other environment without further interactions. Although this problem setting might seem overly ambitious or even far-fetched, in this document we provide the first **tractable formulation of systematic generalization** (Section 3), thanks to a set of structural assumptions that are motivated by a causal viewpoint. Especially, we consider a large, potentially infinite, set of reward-free environments, or a *universe*, the agent can freely interact with. Crucially,

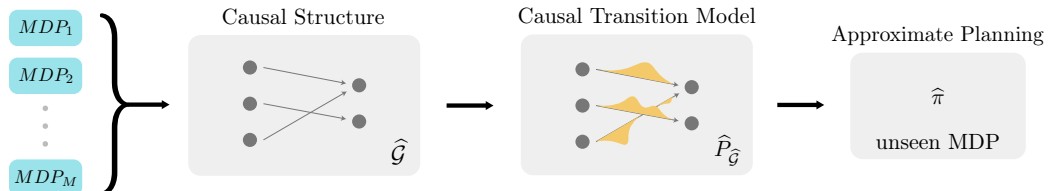

Figure 1: High-level illustration of causal model-based approach to systematic generalization.

these environments share a common causal structure that explains a significant portion, but not all, of their transition dynamics. Can we design a provably efficient algorithm that guarantees an arbitrarily small planning error for any possible task that can be defined over the set of environments, by taking reward-free interactions with a generative model?

In this document, we provide a partially positive answer to this question by presenting a simple but principled causal model-based approach (see Figure 1). This algorithm interacts with a finite subset of the universe to learn the causal structure underlying the set of environments in the form of a causal dependency graph $\mathcal{G}$ Wadhwa & Dong (2021). The causal transition model, which encodes the dynamics that is common across the environment, is obtained by estimating the Bayesian network $P_{\mathcal{G}}$ over $\mathcal{G}$ Dasgupta (1997) from a mixture of the environments. Then, the learned model is employed by a planning oracle to provide an approximately optimal policy for a latent environment and a given reward function. We can show that this simple recipe allows achieving any desired planning error up to an unavoidable error term, which is inherent to the setting. Especially, we provide an **analysis of the sample complexity** (Section 4) of the proposed approach, which is polynomial in all the relevant quantities of the problem.

Finally, with this work we aim to connect several active research areas on reward-free RL Jin et al. (2020), multi-task RL Brunskill & Li (2013), model-based RL Sutton & Barto (2018), factored MDPs Rosenberg & Mansour (2021), causal RL Zhang et al. (2020), experimental design Ghassami et al. (2018), independence testing Canonne et al. (2018), into a general framework where individual progresses can be enhanced beyond the sum of their parts.

## 2 NOTATION

We will denote a set of integers $\{1, \ldots, a\}$ as $[a]$, and the probability simplex over the space $\mathcal{A}$ as $\Delta_{\mathcal{A}}$. For any $A \in \mathcal{A}$, we denote with $A[Z]$ the vector $(A_i)_{i \in Z}$. Given two probability measures $P$ and $Q$ over a discrete space $\mathcal{A}$, their $L_1$-distance is $\|P - Q\|_1 = \sum_{A \in \mathcal{A}} |P(A) - Q(A)|$. We will denote by $\mathcal{U}_{\mathcal{A}}$ the uniform distribution over $\mathcal{A}$.

**Graphs** We define a graph $\mathcal{G}$ as a pair $\mathcal{G} := (\mathcal{V}, E)$, where $\mathcal{V}$ is a set of nodes and $E \subseteq N \times N$ is a set of edges between them. We call $\mathcal{G}$ a *directed graph* if all of its edges $E$ are directed (i.e., ordered pairs of nodes). We also define the in-degree of a node to be its number of incoming edges: $\text{degree}_{\text{in}}(A) = |\{(B, A) : (B, A) \in E, \forall B\}|$. $\mathcal{G}$ is said to be a *Directed Acyclic Graph* (DAG) if it is a directed graph without cycles. We call $\mathcal{G}$ a *bipartite graph* if there exists a partition $X \cup Y = \mathcal{V}$ such that none of the nodes in $X$ and $Y$ are connected by an edge.

**Causal Graphs and Bayesian Networks** For a set $\mathcal{X}$ of random variables, we represent the causal structure over $\mathcal{X}$ with a DAG $\mathcal{G}_{\mathcal{X}} = (\mathcal{X}, E)$,[1] which we call the *causal graph* of $\mathcal{X}$. For each pair of variables $A, B \in \mathcal{X}$, a directed edge $(A, B) \in \mathcal{G}_{\mathcal{X}}$ denotes that $B$ is conditionally dependent on $A$. For every variable $A \in \mathcal{X}$, we denote as $\text{Pa}(A)$ the *causal parents* of $A$, i.e., the set of all the variables $B \in \mathcal{X}$ on which $A$ is conditionally dependent, $(B, A) \in \mathcal{G}_{\mathcal{X}}$. A Bayesian network Dean & Kanazawa (1989) over the set $\mathcal{X}$ is defined as $\mathcal{N} := (\mathcal{G}_{\mathcal{X}}, P)$, where $\mathcal{G}_{\mathcal{X}}$ specifies the *structure* of the network, i.e., the dependencies between the variables in $\mathcal{X}$, and the distribution $P : \mathcal{X} \to \Delta_{\mathcal{X}}$ specifies the conditional probabilities of the variables in $\mathcal{X}$, such that $P(\mathcal{X}) = \prod_{X_i \in \mathcal{X}} P_i(X_i \,|\, \text{Pa}(X_i))$.

**Markov Decision Processes** We define a *discrete* episodic Markov Decision Process (MDP) Puterman (2014) as $\mathcal{M} := ((\mathcal{S}, d_{\mathcal{S}}, n), (\mathcal{A}, d_{\mathcal{A}}, n), P, H, r)$, where $\mathcal{S}$ is a set of $|\mathcal{S}| = S$ states and $\mathcal{A}$ is a set of $|\mathcal{A}| = A$ actions, such that every $s \in \mathcal{S}$ can be represented through a $d_{\mathcal{S}}$-dimensional vector of

---

[1]We will omit the subscript $\mathcal{X}$ whenever clear from the context.

discrete features taking value in $[n]$, and $a \in \mathcal{A}$ through a $d_A$-dimensional vector of discrete features taking value in $[n]$.[2] $P$ is a transition model such that $P(s'|s, a)$ gives the conditional probability distribution of the next state $s'$ having taken action $a$ in state $s$, $H$ is the horizon of an episode, and $r : \mathcal{S} \times \mathcal{A} \to [0, 1]$ is a deterministic reward function. A stochastic *policy* $\pi_h(a|s)$ denotes the conditional probability of taking action $a$ in state $s$ at step $h$. The *value function* $V_h^\pi : \mathcal{S} \to \mathbb{R}$ associated to $\pi$ is defined as $V_h^\pi(s) := \mathbb{E}_\pi \left[ \sum_{h'=h}^H r(s_{h'}, a_{h'}) \mid s_h = s \right]$. We will write $V_{\mathcal{M},r}^\pi$ to denote $V_1^\pi$ in the MDP $\mathcal{M}$ with reward function $r$.

## 3 PROBLEM FORMULATION

In our setting, a learning agent aims to master a large, potentially infinite, set $\mathbb{U}$ of environments modeled as discrete MDPs without rewards,

$$\mathbb{U} := \left\{ \mathcal{M}_i = ((\mathcal{S}, d_S, n), (\mathcal{A}, d_A, n), P_i, \mu) \right\}_{i=1}^\infty,$$

which we call a *universe*. The agent can draw a finite amount of experience by interacting with the MDPs in $\mathbb{U}$. From these interactions alone, the agent aims to acquire sufficient knowledge to approximately solve any task that can be specified over the universe $\mathbb{U}$. Specifically, a *task* is defined as any pairing of an MDP $\mathcal{M} \in \mathbb{U}$ and a reward function $r$, whereas *solving it* refers to providing a slightly sub-optimal policy via planning, i.e., without taking additional interactions. We call this problem *systematic generalization*, which we can formalize as follows.

**Definition 1** (Systematic Generalization). *For any latent MDP $\mathcal{M} \in \mathbb{U}$ and any given reward $r : \mathcal{S} \times \mathcal{A} \to [0, 1]$, the systematic generalization problem requires the agent to provide a policy $\pi$, such that $V_{\mathcal{M},r}^* - V_{\mathcal{M},r}^\pi \leq \epsilon$ up to any desired $\epsilon > 0$.*

Since the set $\mathbb{U}$ is infinite, we clearly require additional structure to make the problem feasible. On the one hand, the state space $(\mathcal{S}, d_S, n)$, action space $(\mathcal{A}, d_A, n)$, and initial state distribution $\mu$ are shared across $\mathcal{M} \in \mathbb{U}$. The transition dynamics $P_i$ is instead specific to each MDP $\mathcal{M}_i \in \mathbb{U}$. However, we assume the presence of a *common causal structure* that underlies the transition dynamics of the universe, and relates the single transition models $P_i$.

### 3.1 THE CAUSAL STRUCTURE OF THE TRANSITION DYNAMICS

To ease the notation, we denote the current state-action features with a random vector $X = (X_i)_{i \in [d_S + d_A]}$, and the next state features with a random vector $Y = (Y_i)_{i \in [d_S]}$. For each environment $\mathcal{M}_i \in \mathbb{U}$, the conditional dependencies between $Y$ and $X$ are represented through a bipartite dependency graph $\mathcal{G}_i$. Clearly, each environment can display its own dependencies, but we assume there is a set of dependencies that represent general causal relationships between the features, and that appear in any $\mathcal{M}_i \in \mathbb{U}$. In particular, we call the intersection $\mathcal{G} := \cap_{i=0}^\infty \mathcal{G}_i$ the *causal structure* of $\mathbb{U}$, which is the set of conditional dependencies that are common across the universe. In Figure 2, we show an illustration of such a causal structure. We assume the causal structure $\mathcal{G}$ is time-consistent,

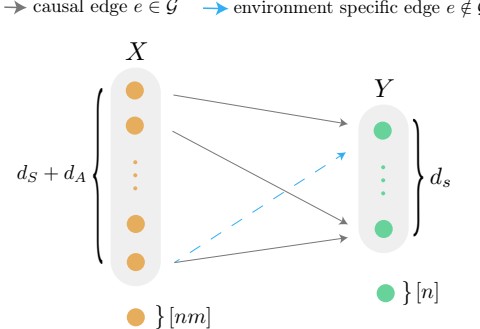

Figure 2: Illustration of the causal transition model $P_\mathcal{G}$.

i.e., $\mathcal{G}^{(h)} = \mathcal{G}^{(1)}$ for any step $h \in [H]$, and sparse, which means that the number of features $X[z]$ on which a feature $Y[j]$ is dependent on is bounded from above.

**Assumption 1** (Z-sparseness). *Let $Z \in \mathbb{N}$. The causal structure $\mathcal{G}$ is $Z$-sparse if $\max_{j \in [d_S]} \text{degree}_{\text{in}}(Y[j]) \leq Z$.*

---

[2] Note that any *tabular* MDP Puterman (2014) can be formulated under this alternative formalism by taking $n = 2$, $d_S = S$, and $d_A = A$, so that the states and actions are specified through one-hot encoding.

Given a causal structure $\mathcal{G}$, and without losing generality,[3] we can express each transition model $P_i$ as

$$P_i(Y|X) = P_{\mathcal{G}}(Y|X)F_i(Y|X), \qquad\qquad P_{\mathcal{G}}(Y|X) = \prod_{j=1}^{d_S} P_j(Y[j]|X[Z_j]),$$

in which $P_{\mathcal{G}}$ is the Bayesian network over the causal structure $\mathcal{G}$, whereas $F_i$ includes environment-specific factors affecting the conditional probabilities,[4] the $Z_j$ are the set of indices $z$ such that $(X[z], Y[j]) \in \mathcal{G}$. Since it represents the conditional probabilities due to universal causal relations in $\mathbb{U}$, we call $P_{\mathcal{G}}$ the *causal transition model* of $\mathbb{U}$. We assume the causal transition model $P_{\mathcal{G}}$ is also time-consistent, i.e., $P_{\mathcal{G}}^{(h)} = P_{\mathcal{G}}^{(1)}, \forall h \in [H]$, and that it explains a significant part of the transition dynamics of $\mathcal{M}_i \in \mathbb{U}$.

**Assumption 2** ($\lambda$-sufficiency). *Let* $\lambda \in [0, 1]$ *be a constant. The causal transition model* $P_{\mathcal{G}}$ *is causally $\lambda$-sufficient if*

$$\sup_X \|P_{\mathcal{G}}(\cdot|X) - P_i(\cdot|X)\|_1 \leq \lambda, \qquad \forall P_i \in \mathcal{M}_i \in \mathbb{U}.$$

Notably, the parameter $\lambda$ controls the amount of the transition dynamics that is due to the universal causal relations $\mathcal{G}$ ($\lambda = 0$ means that $P_{\mathcal{G}}$ is sufficient to explain the transition dynamics of any $\mathcal{M}_i \in \mathbb{U}$, whereas $\lambda = 1$ implies no shared structure between the transition dynamics of the $\mathcal{M}_i \in \mathbb{U}$). In this paper, we argue that learning the causal transition model $P_{\mathcal{G}}$ is a good target for systematic generalization and we provide theoretical support for this claim in Section 4.

### 3.2 A Class of Training Environments

Even if the universe $\mathbb{U}$ admits the structure that we presented in the last section, it is still an infinite set. Instead, the agent can only interact with a finite subset of discrete MDPs

$$\mathbb{M} := \{\mathcal{M}_i = ((\mathcal{S}, d_S, n), (\mathcal{A}, d_A, n), P_i, \mu)\}_{i=1}^M \subset \mathbb{U},$$

which we call a *class* of size $M$. Crucially, the causal structure $\mathcal{G}$ is a property of the full set $\mathbb{U}$, and if we aim to infer it from interactions with a finite class $\mathbb{M}$, we have to assume that $\mathbb{M}$ is informative on the universal causal relations of $\mathbb{U}$.

**Assumption 3** (Diversity). *Let* $\mathbb{M} \subset \mathbb{U}$ *be class of size $M$. We say $\mathbb{M}$ is* causally diverse *if* $\mathcal{G} = \cap_{i=1}^M \mathcal{G}_i = \cap_{i=1}^\infty \mathcal{G}_i$.

Analogously, if we aim to infer the causal transition model $P_{\mathcal{G}}$ from interactions with the transition models $P_i$ of the single MDPs $\mathcal{M}_i \in \mathbb{M}$, we have to assume that $\mathbb{M}$ is balanced in terms of the conditional probabilities displayed by its components, so that the factors that do not represent universal causal relations even out while learning.

**Assumption 4** (Evenness). *Let* $\mathbb{M} \subset \mathbb{U}$ *be class of size $M$. We say $\mathbb{M}$ is* causally even *if* $\mathbb{E}_{i\sim\mathcal{U}_{[M]}}[F_i(Y[j]|X)] = 1, \forall j \in [d_S]$.

Whereas in this paper we assume that $\mathbb{M}$ is *diverse* and *even* by design, we leave as future work the interesting problem of selecting such a class from active interactions with $\mathbb{U}$, which would add to our problem formulation flavors of active learning and experimental design Hauser & Bühlmann (2014); Kocaoglu et al. (2017); Ghassami et al. (2018).

## 4 Sample Complexity of Systematic Generalization with a Generative Model

We have access to a class $\mathbb{M}$ of discrete MDPs within a universe $\mathbb{U}$, from which we can draw interactions with a generative model $P(X)$. We would like to solve the systematic generalization problem as described in Definition 1. This problem requires to provide, for any combination of a (latent) MDP $\mathcal{M} \in \mathbb{U}$, and a given reward function $r$, a planning policy $\widehat{\pi}$ such that $V_{\mathcal{M},r}^* - V_{\mathcal{M},r}^{\widehat{\pi}} \leq \epsilon$. Especially, can we design an algorithm that guarantees this requirement with high probability

---

[3]Note that one can always take $P_{\mathcal{G}}(Y|Z) = 1, \forall(X, Y)$ to avoid shared structure on the transition dynamics.

[4]The parameters in $F_i$ are numerical values such that $P_i$ remains a well-defined probability measure.

---

**Algorithm 1** Causal Transition Model Estimation

---

**Input**: class of MDPs $\mathbb{M}$, error $\epsilon$, confidence $\delta$
let $K' = C'\big(d_S^2 Z^2 n \log(2Md_S^2 d_A/\delta)/\epsilon^2\big)$
set the generative model $P(X) = \mathcal{U}_X$
**for** $i = 1, \ldots, M$ **do**
   let $P_i(Y|X)$ be the transition model of $\mathcal{M}_i \in \mathbb{M}$
   $\widehat{\mathcal{G}}_i \leftarrow$ *Causal Structure Estimation* $(P_i, P(X), K')$
**end for**
let $\widehat{\mathcal{G}} = \cap_{i=1}^M \widehat{\mathcal{G}}_i$
let $K'' = C''\big(d_S^3 n^{3Z+1} \log(4d_S n^Z/\delta)/\epsilon^2\big)$
let $P_{\mathbb{M}}(Y|X)$ be the mixture $\frac{1}{M} \sum_{i=1}^M P_i(Y|X)$
$\widehat{P}_{\widehat{\mathcal{G}}} \leftarrow$ *Bayesian Network Estimation* $(P_{\mathbb{M}}, \widehat{\mathcal{G}}, K'')$
**Output**: causal transition model $\widehat{P}_{\widehat{\mathcal{G}}}$

---

by taking a number of samples $K$ that is polynomial in $\epsilon$ and the relevant parameters of $\mathbb{M}$? Here we give a partially positive answer to this question, by providing a simple but provably efficient algorithm that guarantees systematic generalization over $\mathbb{U}$ up to an unavoidable sub-optimality term $\epsilon_\lambda$ that we will later specify.

The algorithm implements a model-based approach into two separated components. The first, for which we provide the pseudocode in Algorithm 1, is the procedure that actually interacts with the class $\mathbb{M}$ to obtain a principled estimation $\widehat{P}_{\widehat{\mathcal{G}}}$ of the causal transition model $P_{\mathcal{G}}$ of $\mathbb{U}$. The second, is a planning oracle that takes as input a reward function $r$ and the estimated causal transition model, and returns an optimal policy $\widehat{\pi}$ operating on $\widehat{P}_{\widehat{\mathcal{G}}}$ as an approximation of the transition model $P_i$ of the true MDP $\mathcal{M}_i$. We provide an upper bound to the sample complexity of the Algorithm 1.

**Lemma 4.1.** *Let* $\mathbb{M} = \{\mathcal{M}_i\}_{i=1}^M$ *be a class of* $M$ *discrete MDPs, let* $\delta \in (0,1)$, *and let* $\epsilon > 0$. *The Algorithm 1 returns a causal transition model* $\widehat{P}_{\widehat{\mathcal{G}}}$ *such that* $Pr(\|\widehat{P}_{\widehat{\mathcal{G}}} - P_{\mathcal{G}}\|_1 \geq \epsilon) \leq \delta$ *with a sample complexity*

$$K = O\bigg(Md_S^3 Z^2 n^{3Z+1} \log\Big(\frac{4Md_S^2 d_A n^Z}{\delta}\Big) \Big/ \epsilon^2\bigg).$$

Having established the sample complexity of the causal transition model estimation, we can now show how the learned model $\widehat{P}_{\widehat{\mathcal{G}}}$ allows us to approximately solve, via a planning oracle, any task defined by a combination of a latent MDP $\mathcal{M}_i \in \mathbb{U}$ and a given reward function $r$.[5]

**Theorem 4.2.** *Let* $\delta \in (0,1)$ *and* $\epsilon > 0$. *For a latent discrete MDP* $\mathcal{M} \in \mathbb{U}$, *and a given reward function* $r$, *a planning oracle operating on the causal transition model* $\widehat{P}_{\widehat{\mathcal{G}}}$ *as an approximation of* $\mathcal{M}$ *returns a policy* $\widehat{\pi}$ *such that*

$$Pr\big(V_{\mathcal{M}_i,r}^* - V_{\mathcal{M}_i,r} \geq \epsilon_\lambda + \epsilon\big) \leq \delta,$$

*where* $\epsilon_\lambda = 2\lambda H^3 d_S n^{2Z+1}$, *and* $\widehat{P}_{\widehat{\mathcal{G}}}$ *is obtained from Algorithm 1 with* $\delta' = \delta$ *and* $\epsilon' = \epsilon/2H^3 n^{Z+1}$.

Theorem 4.2 establish the sample complexity of systematic generalization through Lemma 4.1. For the discrete MDP setting, we have that $\widetilde{O}(MH^6 d_S^3 Z^2 n^{5Z+3})$, which reduces to $\widetilde{O}(MH^6 S^4 A^2 Z^2)$ in the tabular setting. Unfortunately, we are only able to obtain systematic generalization up to an unavoidable sub-optimality term $\epsilon_\lambda$. This error term is related to the $\lambda$-sufficiency of the causal transition model (Assumption 2), and it accounts for the fact that $P_{\mathcal{G}}$ cannot fully explain the transition dynamics of each $\mathcal{M} \in \mathbb{U}$, even when it is estimated exactly. This is inherent to the ambitious problem setting, and can be only overcome with additional interactions with the test MDP $\mathcal{M}$.

---

[5]To provide this result in the discrete MDP setting, we have to further assume that the transition dynamics $P_i$ of the target MDP $\mathcal{M}_i$ admits factorization analogous to (3.1), such that we can write $P_i(Y|X) = \prod_{j=1}^{d_S} P_{i,j}(Y[j]|X[Z_j'])$, where the scopes $Z_j'$ are given by the environment-specific causal structure $\mathcal{G}_i$, which we assume to be $2Z$-sparse (Assumption 1).

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

# A    RELATED WORK

We revise the relevant literature and we discuss how it relates with our problem formulation and the reported results.

**Reward-Free Reinforcement Learning**    The reward-free RL formulation Jin et al. (2020) is essentially akin to a particular case of our systematic generalization framework (Definition 1) in which the set of MDPs is a singleton $\mathbb{U} = \{\mathcal{M}\}$ instead of an infinite set of MDPs sharing a causal structure. Several recent works have proposed provably efficient algorithms for the reward-free RL formulation, both in tabular Jin et al. (2020); Kaufmann et al. (2021); Ménard et al. (2021); Zhang et al. (2021) and continuous settings with structural assumptions Wang et al. (2020); Zanette et al. (2020); Qiu et al. (2021). It is worth investigating how our sample complexity result compare to an approach that performs reward-free exploration independently for each MDP over a large set $\mathbb{U}$. Let $|\mathbb{U}| = U$, from (Jin et al., 2020, Theorem 4.1) we know that the agnostic reward-free approach would require at least $\Omega(U H S^2 A/\epsilon^2)$ samples to obtain systematic generalization up to an $\epsilon$ threshold over a set of tabular MDPs $\mathbb{U}$. This compares favorably with our $\widetilde{O}(M H^6 S^4 A^2/\epsilon^2)$ complexity (see Corollary B.2) whenever $U$ is small, but leveraging the inner structure of $\mathbb{U}$ becomes more and more important as $U$ grows to infinity, while $M$ remains constant. However, our approach pays this further generality with the additional error term $\epsilon_\lambda$, which is unavoidable. Finally, it is an interesting direction for future work to establish whether the additional factors in $S, A, H$ w.r.t. reward-free RL are also unavoidable.

**Multi-Task Reinforcement Learning**    In the multi-task RL setting Lazaric & Ghavamzadeh (2010); Lazaric & Restelli (2011); Brunskill & Li (2013); Liu et al. (2016), the agent interacts with a set of MDPs (tasks) pursuing generalization from one task to the other, which is in a way similar to the aim of systematic generalization. Although sample complexity results have been derived for the multi-task setting (e.g., Brunskill & Li, 2013), we are not aware of previous works in this stream that considered the problem of interacting with a set of reward-free MDPs, nor the presence of a common causal structure underlying the set of MDPs.

**Model-Based Reinforcement Learning**    The model-based RL Sutton & Barto (2018) methodology prescribes learning an approximate model of transition dynamics in order to learn an optimal policy. Theoretical works (e.g., Jaksch et al., 2010; Ayoub et al., 2020) generally concern with the estimation of the approximate value functions obtained through the learned model, rather than the estimation of the model itself. A notable exception is the work by Tarbouriech et al. (2020), which targets point-wise high probability guarantees on the model estimation as we do in Lemma 4.1, B.1. However, they address the model estimation of a single tabular MDP $\mathcal{M}$, instead of the shared transition dynamics of an infinite set of MDPs $\mathbb{U}$ that we target in this paper. Other works (e.g., Zhang et al., 2020; Tomar et al., 2021) have also addressed model-based RL from a causal perspective. To the best of our knowledge, we are the first to prove a polynomial sample complexity for causal model-based RL in systematic generalization.

**Factored Markov Decision Processes**    The factored MDP formalism Kearns & Koller (1999) allows encoding transition dynamics that are the product of multiple independent factors. This formalism is closely related to how we define the causal transition model as a product of independent factors in (3.1), which can be actually seen as a factored MDP. Several previous works have considered learning problems over factored MDPs, either assuming full knowledge of the underlying factorization structure Delgado et al. (2011); Xu & Tewari (2020); Talebi et al. (2021); Tian et al. (2020), or by estimating the structure from data Strehl et al. (2007); Vigorito & Barto (2009); Chakraborty & Stone (2011); Osband & Van Roy (2014); Rosenberg & Mansour (2021). To the best of our knowledge, none of the existing works have considered the factored MDP framework in combination with a reward-free and multiple-environment setting and systematic generalization, which bring unique challenges to the identification of the underlying factorization and the estimation of the transition factors.

# B    ANCILLARY RESULTS

We provide some additional results w.r.t. the sample complexity analysis we reported in Section 4. First, we reduce the reported sample complexity of systematic generalization for tabular domains.

Then, we provide sample complexity results on the estimation of the causal structure (Section B.1) and the Bayesian network (Section B.2) of an MDP, which can be of independent interest.

We can derive an analogous result to Lemma 4.1 for the tabular MDP setting, as stated by the following lemma.

**Lemma B.1.** *Let* $\mathbb{M} = \{\mathcal{M}_i\}_{i=1}^M$ *be a class of* $M$ *tabular MDPs. The sample complexity of Lemma 4.1 reduces to*

$$K = O\left( \frac{MS^2 Z^2 2^{2Z} \log\left(\frac{4MS^2 A 2^Z}{\delta}\right)}{\epsilon^2} \right).$$

Then, the result of Theorem 4.2 reduces to the analogous for the tabular MDP setting without requiring any additional factorization of the environment-specific transition model.

**Corollary B.2.** *For a tabular MDP* $\mathcal{M} \in \mathbb{M}$, *the result of Theorem 4.2 holds with* $\epsilon_\lambda = 2\lambda SAH^3$, $\epsilon' = \epsilon/2SAH^3$.

### B.1 SAMPLE COMPLEXITY OF LEARNING THE CAUSAL STRUCTURE OF A DISCRETE MDP

As a byproduct of the main result in Theorem 4.2, we can provide a specific sample complexity result for the problem of learning the causal structure $\mathcal{G}$ underlying a discrete MDP $\mathcal{M}$ with a generative model. We believe that this problem can be of independent interest, mainly in consideration of previous work on causal discovery of general stochastic processes (e.g., Wadhwa & Dong, 2021), for which we refine known results to account for the structure of an MDP, which allows for a tighter analysis of the sample complexity.

Instead of targeting the exact dependency graph $\mathcal{G}$, which can include dependencies that are so weak to be nearly impossible to detect with a finite number of samples, we only address the dependencies above a given $\epsilon$ threshold.

**Definition 2.** *We call* $\mathcal{G}_\epsilon \subseteq \mathcal{G}$ *the* $\epsilon$-*dependency subgraph of* $\mathcal{G}$ *if it holds, for each pair* $(A, B) \in \mathcal{G}$ *distributed as* $P_{A,B}$

$$(A, B) \in \mathcal{G}_\epsilon \quad iff \quad \inf_{Q \in \{\Delta_A \times \Delta_B\}} \|P_{A,B} - Q\|_1 \geq \epsilon.$$

Before presenting the sample complexity result, we state the existence of a principled independence testing procedure.

**Lemma B.3** (Diakonikolas et al. (2021)). *There exists an* $(\epsilon, \delta)$-*independence tester* $\mathbb{I}(A, B)$ *for distributions* $P_{A,B}$ *on* $[n] \times [n]$, *which returns with probability at least* $1 - \delta$

- *yes, if* $A, B$ *are independent,*
- *no, if* $\inf_{Q \in \{\Delta_A \times \Delta_B\}} \|P_{A,B} - Q\|_1 \geq \epsilon$,

*with a sample complexity* $O(n \log(1/\delta)/\epsilon^2)$.

We can now provide an upper bound to the number of samples required by a simple estimation procedure to return an $(\epsilon, \delta)$-estimate $\widehat{\mathcal{G}}$ of the causal dependency graph $\mathcal{G}$.

**Theorem B.4.** *Let* $\mathcal{M}$ *be a discrete MDP with an underlying causal structure* $\mathcal{G}$, *let* $\delta \in (0, 1)$, *and let* $\epsilon > 0$. *The Algorithm 2 returns a dependency graph* $\widehat{\mathcal{G}}$ *such that* $Pr(\widehat{\mathcal{G}} \neq \mathcal{G}_\epsilon) \leq \delta$ *with a sample complexity*

$$K = O\big(n \log(d_S^2 d_A/\delta)/\epsilon^2\big).$$

Finally, we can state an analogous result for a tabular MDP setting, by taking $n = 2, d_S = S, d_A = A$.

**Corollary B.5.** *Let* $\mathcal{M}$ *be a tabular MDP. The result of Theorem B.4 reduces to* $K = O\big(\log(S^2 A/\delta)/\epsilon^2\big)$.

### B.2 SAMPLE COMPLEXITY OF LEARNING THE BAYESIAN NETWORK OF A DISCRETE MDP

We present as a standalone result an upper bound to the sample complexity of learning the parameters of a Bayesian network $P_{\mathcal{G}}$ with a fixed structure $\mathcal{G}$. Especially, we refine known results (e.g.,

---

**Algorithm 2** Causal Structure Estimation for an MDP

---

**Input**: sampling model $P(Y|X)$, generative model $P(X)$, batch parameter $K$
draw $(x_k, y_k)_{k=1}^K \overset{\text{iid}}{\sim} P(Y|X)P(X)$
initialize $\widehat{\mathcal{G}} = \emptyset$
**for** each pair of nodes $X_z, Y_j$ **do**
    compute the independence test $\mathbb{I}(X_z, Y_j)$
    if a dependency is found add $(X_z, Y_j)$ to $\widehat{\mathcal{G}}$
**end for**
**Output**: causal dependency graph $\widehat{\mathcal{G}}$

---

---

**Algorithm 3** Bayesian Network Estimation for an MDP

---

**Input**: sampling model $P(Y|X)$, dependency graph $\mathcal{G}$, batch parameter $K$
let $K' = \lceil K/d_S n^Z \rceil$
**for** $j = 1, \ldots, d_S$ **do**
    let $Z_j$ the scopes $(X[Z_j], Y[j]) \subseteq \mathcal{G}$
    initialize the counts $N(X[Z_j], Y[j]) = 0$
    **for** each value $x \in [n]^{|Z_j|}$ **do**
        **for** $k = 1, \ldots, K'$ **do**
            draw $y \sim P(Y[j]|X[Z_j] = x)$
            increment $N(X[Z_j] = x, Y[j] = y)$
        **end for**
    **end for**
    compute $\widehat{P}_j(Y[j]|X[Z_j]) = N(X[Z_j], Y[j])/K'$
**end for**
let $\widehat{P}_{\mathcal{G}}(Y|X) = \prod_{j=1}^{d_S} \widehat{P}_j(Y[j]|X[Z_j])$
**Output**: Bayesian network $\widehat{P}_{\mathcal{G}}$

---

Friedman & Yakhini, 1996; Dasgupta, 1997; Cheng et al., 2002; Abbeel et al., 2006; Canonne et al., 2017) by considering the specific structure $\mathcal{G}$ of an MDP.

If the structure $\mathcal{G}$ is dense, the number of parameters of $P_{\mathcal{G}}$ grows exponentially, making the estimation problem mostly intractable. Thus, we consider a $Z$-sparse $\mathcal{G}$ (Assumption 1), as in previous works Dasgupta (1997). Then, we can provide a polynomial sample complexity for the problem of learning the Bayesian network $P_{\mathcal{G}}$ of a discrete MDP $\mathcal{M}$.

**Theorem B.6.** *Let $\mathcal{M}$ be a discrete MDP, let $\mathcal{G}$ be its underlying causal structure, let $\delta \in (0, 1)$, and let $\epsilon > 0$. The Algorithm 3 returns a Bayesian network $\widehat{P}_{\mathcal{G}}$ such that $Pr(\|\widehat{P}_{\mathcal{G}} - P_{\mathcal{G}}\|_1 \geq \epsilon) \leq \delta$ with a sample complexity*

$$K = O\big(d_S^3 n^{3Z+1} \log(d_S n^Z/\delta)/\epsilon^2\big).$$

Analogously, we can state the sample complexity of learning the fixed structure Bayesian network of a tabular MDP.

**Corollary B.7.** *Let $\mathcal{M}$ be a tabular MDP. The result of Theorem B.6 reduces to $K = O\big(S^2 2^{2Z} \log(S 2^Z/\delta)/\epsilon^2\big)$.*

## C PROOFS

**Proposition 1.** *The causal structure $\mathcal{G}$ of $\mathbb{U}$ can be identified from purely observational data.*

*Proof.* First, recall that with observational data alone, a causal graph can be identified up to its Markov equivalence class Hauser & Bühlmann (2014). This means that its skeleton and v-structure are properly identified, meanwhile determining the edge orientations requires interventional data in the general case. Since in the considered causal graph $\mathcal{G}$ the edges orientations are determined a priori (as they follow the direction of time), the causal graph can be entirely determined by using only observational data. $\square$

PROOFS OF SECTION 4: CAUSAL TRANSITION MODEL ESTIMATION

Before reporting the proof of the main result in Theorem 4.2, it is worth considering a set of lemmas that will be instrumental to the main proof.

First, we provide an upper bound to the L1-norm between the Bayesian network $P_\mathcal{G}$ over a given structure $\mathcal{G}$ and the Bayesian network $P_{\mathcal{G}_\epsilon}$ over the structure $\mathcal{G}_\epsilon$, which is the $\epsilon$-dependency subgraph of $\mathcal{G}$ as defined in Definition 2.

**Lemma C.1.** *Let $\mathcal{G}$ a Z-sparse dependency graph, and let $\mathcal{G}_\epsilon$ its corresponding $\epsilon$-dependence subgraph for a threshold $\epsilon > 0$. The L1-norm between the Bayesian network $P_\mathcal{G}$ over $\mathcal{G}$ and the Bayesian network $P_{\mathcal{G}_\epsilon}$ over $\mathcal{G}_\epsilon$ can be upper bounded as*

$$\|P_\mathcal{G} - P_{\mathcal{G}_\epsilon}\|_1 \le d_S Z \epsilon.$$

*Proof.* The proof is based on the fact that every edge $(X_i, Y_j)$ such that $(X_i, Y_j) \in \mathcal{G}$ and $(X_i, Y_j) \notin \mathcal{G}_\epsilon$ corresponds to a weak conditional dependence (see Definition 2), which means that $\|P_{Y_j|X_i} - P_{Y_j}\|_1 \le \epsilon$.

We denote with $Z_j$ the scopes of the parents of the node $Y[j]$ in $\mathcal{G}$, i.e., $\mathrm{Pa}_\mathcal{G}(Y[j]) = X[Z_j]$, and with $Z_{j,\epsilon}$ the scopes of the parents of the node $Y[j]$ in $\mathcal{G}_\epsilon$, i.e., $\mathrm{Pa}_{\mathcal{G}_\epsilon}(Y[j]) = X[Z_{j,\epsilon}]$. As a direct consequence of Definition 2, we have $Z_{j,\epsilon} \subseteq Z_j$ for any $j \in d_S$, and we can write

$$P_\mathcal{G}(Y|X) = \prod_{j=1}^{d_S} P_j(Y[j] \mid X[Z_j]) = \prod_{j=1}^{d_S} P_j(Y[j] \mid X[Z_{j,\epsilon}], X[Z_j \backslash Z_{j,\epsilon}]), \qquad P_{\mathcal{G}_\epsilon}(Y|X) = \prod_{j=1}^{d_S} P_j(Y[j] \mid X[Z_{j,\epsilon}]).$$

Then, we let $Z_j \setminus Z_{j,\epsilon} = [I]$ overwriting the actual indices for the sake of clarity, and we derive

$$\|P_\mathcal{G} - P_{\mathcal{G}_\epsilon}\|_1 \le \sum_{j=1}^{d_S} \left\| P_j(Y[j] \mid X[Z_{j,\epsilon}], \cup_{i=1}^I X[i]) - P_j(Y[j] \mid X[Z_{j,\epsilon}]) \right\|_1 \tag{1}$$

$$\le \sum_{j=1}^{d_S} \sum_{i'=1}^{I} \left\| P_j(Y[j] \mid X[Z_{j,\epsilon}], \cup_{i=i'}^I X[i]) - P_j(Y[j] \mid X[Z_{j,\epsilon}], \cup_{i=i'+1}^I X[i]) \right\|_1 \tag{2}$$

$$\le \sum_{j=1}^{d_S} \sum_{i'=1}^{I} \epsilon \le d_S Z \epsilon, \tag{3}$$

in which we employed the property $\|\mu - \nu\|_1 \le \|\prod_i \mu_i - \prod_i \nu_i\|_1 \le \sum_i \|\mu_i - \nu_i\|_1$ for the L1-norm between product distributions $\mu = \prod_i \mu_i, \nu = \prod_i \nu_i$ to write (1), we repeatedly applied the triangle inequality $\|\mu - \nu\|_1 \le \|\mu - \rho\|_1 + \|\rho - \nu\|_1$ to get (2) from (1), we upper bounded each term of the sum in (2) with $\epsilon$ thanks to Definition 2, and we finally employed the Z-sparseness Assumption 1 to upper bound $I$ with $Z$ in (3). $\square$

Next, we provide a crucial sample complexity result for a provably efficient estimation of a Bayesian network $\widehat{P}_{\widehat{\mathcal{G}}}$ over an estimated $\epsilon$-dependency subgraph $\widehat{\mathcal{G}}$, which relies on both the causal structure estimation result of Theorem B.4 and the Bayesian network estimation result of Theorem B.6.

**Lemma C.2.** *Let $\mathcal{M}$ be a discrete MDP, let $\mathbb{M} = \{\mathcal{M}\}$ be a singleton class, let $\delta \in (0,1)$, and let $\epsilon > 0$. The Algorithm 1 returns a Bayesian network $\widehat{P}_{\widehat{\mathcal{G}}}$ such that $Pr(\|\widehat{P}_{\widehat{\mathcal{G}}} - P_\mathcal{G}\|_1 \ge \epsilon) \le \delta$ with a sample complexity*

$$K = O\left( \frac{d_S^3 \, Z^2 \, n^{3Z+1} \, \log\left(\frac{4 d_S^2 d_A n^Z}{\delta}\right)}{\epsilon^2} \right).$$

*Proof.* We aim to obtain the number of samples $K = K' + K''$ for which Algorithm 1 is guaranteed to return a Bayesian network estimate $\widehat{P}_{\widehat{\mathcal{G}}}$ over a causal structure estimate $\widehat{\mathcal{G}}$ such that $Pr(\|\widehat{P}_{\widehat{\mathcal{G}}} - P_\mathcal{G}\|_1 \ge \epsilon) \le \delta$ in a setting with a singleton class of discrete MDPs. First, we derive the following decomposition of the error

$$\|\widehat{P}_{\widehat{\mathcal{G}}} - P_\mathcal{G}\|_1 \le \|\widehat{P}_{\widehat{\mathcal{G}}} \pm P_{\widehat{\mathcal{G}}} \pm P_{\mathcal{G}_{\epsilon'}} - P_\mathcal{G}\|_1 \le \|\widehat{P}_{\widehat{\mathcal{G}}} - P_{\widehat{\mathcal{G}}}\|_1 + \|P_{\widehat{\mathcal{G}}} - P_{\mathcal{G}_{\epsilon'}}\|_1 + \|P_{\mathcal{G}_{\epsilon'}} - P_\mathcal{G}\|_1 \tag{4}$$

in which we employed the triangle inequality $\|\mu - \nu\|_1 \leq \|\mu - \rho\|_1 + \|\rho - \nu\|_1$. Then, we can write

$$Pr\big(\|\widehat{P}_{\widehat{\mathcal{G}}} - P_{\mathcal{G}}\|_1 \geq \epsilon\big) \leq \underbrace{Pr\Big(\|\widehat{P}_{\widehat{\mathcal{G}}} - P_{\widehat{\mathcal{G}}}\|_1 \geq \frac{\epsilon}{3}\Big)}_{\text{Bayesian network estimation } (\star)} + \underbrace{Pr\Big(\|P_{\widehat{\mathcal{G}}} - P_{\mathcal{G}_{\epsilon'}}\|_1 \geq \frac{\epsilon}{3}\Big)}_{\text{causal structure estimation } (\bullet)} + \underbrace{Pr\Big(\|P_{\mathcal{G}_{\epsilon'}} - P_{\mathcal{G}}\|_1 \geq \frac{\epsilon}{3}\Big)}_{\text{Bayesian network subgraph } (\diamond)}$$

through the decomposition (4) and a union bound to isolate the three independent sources of error $(\star), (\bullet), (\diamond)$. To upper bound the latter term $(\diamond)$ with 0, we invoke Lemma C.1 to have $d_s Z \epsilon' \leq \frac{\epsilon}{3}$, which gives $\epsilon' \leq \frac{\epsilon}{3 d_S Z}$. Then, we consider the middle term $(\bullet)$, for which we can write

$$Pr\Big(\|P_{\widehat{\mathcal{G}}_{\epsilon'}} - P_{\mathcal{G}_{\epsilon'}}\|_1 \geq \frac{\epsilon}{3}\Big) \leq Pr\big(\widehat{\mathcal{G}} \neq \mathcal{G}_\epsilon\big). \tag{5}$$

We can now upper bound $(\bullet) \leq \delta/2$ through (5) by invoking Theorem B.4 with threshold $\epsilon' = \frac{\epsilon}{3 d_S Z}$ and confidence $\delta' = \frac{\delta}{2}$, which gives

$$K' = C'\bigg(\frac{d_S^{4/3} \ Z^{4/3} \ n \ \log^{1/3}(2 d_S^2 d_A/\delta)}{\epsilon^{4/3}} + \frac{d_S^2 \ Z^2 \ n \ \log^{1/2}(2 d_S^2 d_A/\delta) + \log(2 d_S^2 d_A/\delta)}{\epsilon^2}\bigg). \tag{6}$$

Next, we can upper bound $(\star) \leq \delta/2$ by invoking Theorem B.6 with threshold $\epsilon' = \frac{\epsilon}{3}$ and confidence $\delta' = \frac{\delta}{2}$, which gives

$$K'' = C''\bigg(\frac{d_S^3 \ n^{3Z+1} \ \log(4 d_S n^Z/\delta)}{\epsilon^2}\bigg). \tag{7}$$

Finally, through the combination of (6) and (7), we can derive the sample complexity that guarantees $Pr(\|\widehat{P}_{\widehat{\mathcal{G}}} - P_{\mathcal{G}}\|_1 \geq \epsilon) \leq \delta$ under the assumption $\epsilon^{4/3} \ll \epsilon^2$, i.e.,

$$K = K' + K'' \leq \frac{d_S^3 \ Z^2 \ n^{3Z+1} \ \log\Big(\frac{4 d_S^2 d_A n^Z}{\delta}\Big)}{\epsilon^2},$$

which concludes the proof. $\qquad\square$

Whereas Lemma C.2 is concerned with the sample complexity of learning the Bayesian network of a singleton class, we can now extend the result to account for a class $\mathbb{M}$ composed of $M$ discrete MDPs.

**Lemma 4.1.** *Let $\mathbb{M} = \{\mathcal{M}_i\}_{i=1}^M$ be a class of $M$ discrete MDPs, let $\delta \in (0,1)$, and let $\epsilon > 0$. The Algorithm 1 returns a causal transition model $\widehat{P}_{\widehat{\mathcal{G}}}$ such that $Pr(\|\widehat{P}_{\widehat{\mathcal{G}}} - P_{\mathcal{G}}\|_1 \geq \epsilon) \leq \delta$ with a sample complexity*

$$K = O\bigg(M d_S^3 Z^2 n^{3Z+1} \log\Big(\frac{4 M d_S^2 d_A n^Z}{\delta}\Big) \Big/ \epsilon^2\bigg).$$

*Proof.* We aim to obtain the number of samples $K = M K' + K''$ for which Algorithm 1 is guaranteed to return a Bayesian network estimate $\widehat{P}_{\widehat{\mathcal{G}}}$ over a causal structure estimate $\widehat{\mathcal{G}}$ such that $Pr(\|\widehat{P}_{\widehat{\mathcal{G}}} - P_{\mathcal{G}}\|_1 \geq \epsilon) \leq \delta$ in a setting with a class of $M$ discrete MDPs. First, we can derive an analogous decomposition as in (4), such that we have

$$Pr\big(\|\widehat{P}_{\widehat{\mathcal{G}}} - P_{\mathcal{G}}\|_1 \geq \epsilon\big) \leq \underbrace{Pr\Big(\|\widehat{P}_{\widehat{\mathcal{G}}} - P_{\widehat{\mathcal{G}}}\|_1 \geq \frac{\epsilon}{3}\Big)}_{\text{Bayesian network estimation } (\star)} + \underbrace{Pr\Big(\|P_{\widehat{\mathcal{G}}} - P_{\mathcal{G}_{\epsilon'}}\|_1 \geq \frac{\epsilon}{3}\Big)}_{\text{causal structure estimation } (\bullet)} + \underbrace{Pr\Big(\|P_{\mathcal{G}_{\epsilon'}} - P_{\mathcal{G}}\|_1 \geq \frac{\epsilon}{3}\Big)}_{\text{Bayesian network subgraph } (\diamond)}$$

through a union bound. Crucially, the terms $(\star), (\diamond)$ are unaffected by the class size, which leads to $K'' = (7)$ by upper bounding $(\star)$, and $\epsilon' \leq \frac{\epsilon}{3 d_S Z}$ by upper bounding $(\diamond)$, exactly as in the proof of Lemma C.2. Instead, the number of samples $K'$ has to guarantee that $(\bullet) = Pr(\|P_{\widehat{\mathcal{G}}} - P_{\mathcal{G}_{\epsilon'}}\|_1 \geq \epsilon/3) \leq \delta/2$, where the causal structure $\mathcal{G}_{\epsilon'}$ is now the intersection of the causal structures of the single class components $\mathcal{M}_i$, i.e., $\mathcal{G}_{\epsilon'} = \cap_{i=1}^M \mathcal{G}_{\epsilon',i}$. Especially, we can write

$$(\bullet) = Pr\Big(\|P_{\widehat{\mathcal{G}}} - P_{\mathcal{G}_{\epsilon'}}\|_1 \geq \frac{\epsilon}{3}\Big) \leq Pr\big(\widehat{\mathcal{G}} \neq \mathcal{G}_{\epsilon'}\big) \leq Pr\bigg(\bigcup_{i=1}^M \widehat{\mathcal{G}}_i \neq \mathcal{G}_{\epsilon',i}\bigg) \leq \sum_{i=0}^M Pr\big(\widehat{\mathcal{G}}_i \neq \mathcal{G}_{\epsilon',i}\big), \tag{8}$$

through a union bound on the estimation of the single causal structures $\widehat{\mathcal{G}}_i$. Then, we can upper bound $(\bullet) \leq \delta/2$ through (8) by invoking Theorem B.4 with threshold $\epsilon' = \frac{\epsilon}{3d_S Z}$ and confidence $\delta' = \frac{\delta}{2M}$, which gives

$$K' = C' \left( \frac{d_S^{4/3} Z^{4/3} n \log^{1/3}(2Md_S^2 d_A/\delta)}{\epsilon^{4/3}} + \frac{d_S^2 Z^2 n \log^{1/2}(2Md_S^2 d_A/\delta) + \log(2Md_S^2 d_A/\delta)}{\epsilon^2} \right).$$

(9)

Finally, through the combination of (9) and (7), we can derive the sample complexity that guarantees $Pr(\|\widehat{P}_{\widehat{\mathcal{G}}} - P_{\mathcal{G}}\|_1 \geq \epsilon) \leq \delta$ under the assumption $\epsilon^{4/3} \ll \epsilon^2$, i.e.,

$$K = MK' + K'' \leq \frac{Md_S^3 Z^2 n^{3Z+1} \log\left(\frac{4Md_S^2 d_A n^Z}{\delta}\right)}{\epsilon^2},$$

which concludes the proof. $\qquad\square$

It is now straightforward to extend Lemma 4.1 for a class $\mathbb{M}$ composed of $M$ tabular MDPs.

**Lemma B.1.** *Let $\mathbb{M} = \{\mathcal{M}_i\}_{i=1}^M$ be a class of $M$ tabular MDPs. The sample complexity of Lemma 4.1 reduces to*

$$K = O\left( \frac{MS^2 Z^2 2^{2Z} \log\left(\frac{4MS^2 A 2^Z}{\delta}\right)}{\epsilon^2} \right).$$

*Proof.* To obtain $K = MK' + K''$, we follows similar steps as in the proof of Lemma 4.1, to have the usual decomposition of the event $Pr(\|\widehat{P}_{\widehat{\mathcal{G}}} - P_{\mathcal{G}}\|_1 \geq \epsilon)$ in the $(\star), (\bullet), (\diamond)$ terms. We can deal with $(\diamond)$ as in Lemma 4.1 to get $\epsilon' \leq \frac{\epsilon}{3SZ}$. Then, we upper bound $(\bullet) \leq \delta/2$ by invoking Corollary B.5 (instead of Theorem B.4) with threshold $\epsilon' = \frac{\epsilon}{3SZ}$ and confidence $\delta' = \frac{\delta}{2M}$, which gives

$$K' = C'\left( \frac{S^{4/3} Z^{4/3} \log^{1/3}(2MS^2 A/\delta)}{\epsilon^{4/3}} + \frac{S^2 Z^2 \log^{1/2}(2MS^2 A/\delta) + \log(2MS^2 A/\delta)}{\epsilon^2} \right).$$

(10)

Similarly, we upper bound $(\star) \leq \delta/2$ by invoking Corollary B.7 (instead of Theorem B.3) with threshold $\epsilon' = \frac{\epsilon}{3}$ and confidence $\delta' = \frac{\delta}{2}$, which gives

$$K'' = \frac{18 S^2 2^{2Z} \log(4S2^Z/\delta)}{\epsilon^2}.$$

(11)

Finally, we combine 10 with 11 to obtain

$$K = MK' + K'' \leq \frac{M S^2 Z^2 2^{2Z} \log\left(\frac{4MS^2 A 2^Z}{\delta}\right)}{\epsilon^2}$$

$\qquad\square$

PROOFS OF SECTION 4: PLANNING

For the sake of notational clarity within the following proofs we express

**Theorem 4.2.** *Let $\delta \in (0,1)$ and $\epsilon > 0$. For a latent discrete MDP $\mathcal{M} \in \mathbb{U}$, and a given reward function $r$, a planning oracle operating on the causal transition model $\widehat{P}_{\widehat{\mathcal{G}}}$ as an approximation of $\mathcal{M}$ returns a policy $\widehat{\pi}$ such that*

$$Pr\left( V^*_{\mathcal{M}_i, r} - V_{\mathcal{M}_i, r} \geq \epsilon_\lambda + \epsilon \right) \leq \delta,$$

*where $\epsilon_\lambda = 2\lambda H^3 d_S n^{2Z+1}$, and $\widehat{P}_{\widehat{\mathcal{G}}}$ is obtained from Algorithm 1 with $\delta' = \delta$ and $\epsilon' = \epsilon/2H^3 n^{Z+1}$.*

*Proof.* Consider the MDPs with transition model $P$ and $\widehat{P}_{\widehat{\mathcal{G}}}$. We refer to the respective optimal policies as $\pi^*$ and $\widehat{\pi}^*$. Moreover, since the reward $r$ is fixed, we remove it from the expressions for the sake of clarity, and refer with $\widehat{V}$ to the value function of the MDP with transition model $\widehat{P}_{\widehat{\mathcal{G}}}$. As done in (Jin et al., 2020, Theorem 3.5), we can write the following decomposition, where $V^* := V^{\pi^*}$.

$$
\mathbb{E}_{s_1 \sim P}\left[ V_1^*(s_1) - V_1^{\widehat{\pi}}(s_1) \right] \leq \underbrace{\left\lfloor \mathbb{E}_{s_1 \sim P}\left[ V_1^*(s_1) - \widehat{V}_1^{\widehat{\pi}^*}(s_1) \right] \right\rfloor}_{\text{evaluation error}} + \underbrace{\mathbb{E}_{s_1 \sim P}\left[ \widehat{V}_1^*(s_1) - \widehat{V}_1^{\widehat{\pi}^*}(s_1) \right]}_{\leq\ 0 \text{ by def.}}
$$

$$
+ \underbrace{\mathbb{E}_{s_1 \sim P}\left[ \widehat{V}_1^{\widehat{\pi}^*}(s_1) - \widehat{V}_1^{\widehat{\pi}}(s_1) \right]}_{\text{optimization error}} + \underbrace{\left\lfloor \mathbb{E}_{s_1 \sim P}\left[ \widehat{V}_1^{\widehat{\pi}}(s_1) - V_1^{\widehat{\pi}}(s_1) \right] \right\rfloor}_{\text{evaluation error}}
$$

$$
\leq \underbrace{2n^{Z+1} H^3 \epsilon'}_{\epsilon} + \underbrace{2n^{2Z+1} d_S H^3 \lambda}_{\epsilon_\lambda}
$$

where in the last step we have set to 0 the approximation due to the planning oracle assumption, and we have bounded the evaluation errors according to Lemma C.3. In order to get $2n^{Z+1} H^3 \epsilon' = \epsilon$ we have to set $\epsilon' = \frac{\epsilon}{2n^{Z+1} H^3}$. Considering the sample complexity result in Lemma 4.1 the final sample complexity will be:

$$
K = O\left( \frac{M\ d_S^3\ Z^2\ n^{3Z+1}\ \log\left( \frac{4M d_S^2 d_A n^Z}{\delta} \right)}{(\epsilon')^2} \right) = O\left( \frac{4\ M\ d_S^3\ Z^2\ n^{5Z+3}\ H^6\ \log\left( \frac{4M d_S^2 d_A n^Z}{\delta} \right)}{\epsilon^2} \right)
$$

$\square$

**Lemma C.3.** *Under the preconditions of Theorem 4.2, with probability $1 - \delta$, for any reward function $r$ and policy $\pi$, we can bound the value function estimation error as follows.*

$$
\left| \mathbb{E}_{s \sim P}\left[ \widehat{V}_{1,r}^{\pi}(s) - V_{1,r}^{\pi}(s) \right] \right| \leq \underbrace{n^{Z+1} H^3 \epsilon'}_{\epsilon} + \underbrace{n^{2Z+1} d_S H^3 \lambda}_{\epsilon_\lambda} \tag{12}
$$

*where $\widehat{V}$ is the value function of the MDP with transition model $\widehat{P}_{\widehat{\mathcal{G}}}$, $\epsilon'$ is the approximation error between $\widehat{P}_{\widehat{\mathcal{G}}}$ and $P_G$ studied in Lemma 4.1, and $\lambda$ stands for the $\lambda$-sufficiency parameter of $P_{\mathcal{G}}$.*

*Proof.* The proof will be along the lines of that of Lemma 3.6 in (Jin et al., 2020). We first recall (Dann et al., 2017, Lemma E.15), which we restate in Lemma C.5. In this prof we consider an environment specific true MDP $\mathcal{M}$ with transition model $P$, and an mdp $\widehat{\mathcal{M}}$ that has as transition model the estimated causal transition model $\widehat{P}_{\widehat{\mathcal{G}}}$. In the following, the expectations will be w.r.t. $P$. Moreover, since the reward $r$ is fixed, we remove it from the expressions for the sake of clarity. We can start deriving

$$
\left| \mathbb{E}_{s \sim P}\left[ \widehat{V}_1^{\pi}(s) - V_1^{\pi}(s) \right] \right| \leq \left| \mathbb{E}_X\left[ \sum_{h=1}^{H} (\widehat{P}_{\widehat{\mathcal{G}}} - P) \widehat{V}_{h+1}^{\pi}(X) \right] \right|
$$

$$
\leq \mathbb{E}_X\left[ \sum_{h=1}^{H} \left| (\widehat{P}_{\widehat{\mathcal{G}}} - P) \widehat{V}_{h+1}^{\pi}(X) \right| \right]
$$

$$
= \sum_{h=1}^{H} \mathbb{E}_X \left| (\widehat{P}_{\widehat{\mathcal{G}}} - P) \widehat{V}_{h+1}^{\pi}(X) \right| \tag{13}
$$

We now bound a single term within the sum above as follows:

$$
\mathbb{E}_X \left| (\widehat{P}_{\widehat{\mathcal{G}}} - P) \widehat{V}_{h+1}^{\pi}(X) \right| = \mathbb{E}_X \left| (\widehat{P}_{\widehat{\mathcal{G}}} - P_{\mathcal{G}} + P_{\mathcal{G}} - P) \widehat{V}_{h+1}^{\pi}(X) \right|
$$

$$
= \mathbb{E}_X \left| (\widehat{P}_{\widehat{\mathcal{G}}} - P_{\mathcal{G}}) \widehat{V}_{h+1}^{\pi}(X) + (P_{\mathcal{G}} - P) \widehat{V}_{h+1}^{\pi}(X) \right|
$$

$$
\leq \mathbb{E}_X \left[ \left| (\widehat{P}_{\widehat{\mathcal{G}}} - P_{\mathcal{G}}) \widehat{V}_{h+1}^{\pi}(X) \right| + \left| (P_{\mathcal{G}} - P) \widehat{V}_{h+1}^{\pi}(X) \right| \right]
$$

$$
= \mathbb{E}_X \left| (\widehat{P}_{\widehat{\mathcal{G}}} - P_{\mathcal{G}}) \widehat{V}_{h+1}^{\pi}(X) \right| + \mathbb{E}_X \left| (P_{\mathcal{G}} - P) \widehat{V}_{h+1}^{\pi}(X) \right| \tag{14}
$$

We can now bound each term. Let us start considering the first term:

$$
\begin{aligned}
\mathbb{E}_X \left| (\widehat{P}_{\widehat{\mathcal{G}}} - P_{\mathcal{G}}) \widehat{V}_{h+1}^\pi (X) \right| &= \mathbb{E}_X \left| \widehat{P}_{\widehat{\mathcal{G}}} \widehat{V}_{h+1}^\pi (X) - P_{\mathcal{G}} \widehat{V}_{h+1}^\pi (X) \right| \\
&= \mathbb{E}_X \left| \sum_Y \widehat{P}_{\widehat{\mathcal{G}}}(Y|X) \widehat{V}_{h+1}^\pi(Y) - \sum_Y P_{\mathcal{G}}(Y|X) \widehat{V}_{h+1}^\pi(Y) \right| \\
&= \mathbb{E}_X \left| \sum_Y \widehat{P}_{\widehat{\mathcal{G}}}(Y|X) \, \mathbb{E}_{X'\sim\pi} \left[ r(X') + \widehat{P}_{\widehat{\mathcal{G}}} \widehat{V}_{h+2}^\pi(X') \right] \right. \\
&\quad \left. - \sum_Y P_{\mathcal{G}}(Y|X) \, \mathbb{E}_{X'\sim\pi} \left[ r(X') + P_{\mathcal{G}} \widehat{V}_{h+2}^\pi(X') \right] \right| \\
&= \mathbb{E}_X \left| \sum_Y \left( \widehat{P}_{\widehat{\mathcal{G}}}(Y|X) - P_{\mathcal{G}}(Y|X) \right) \mathbb{E}_{X'\sim\pi} \left[ r(X') \right] \right. \\
&\quad \left. + \sum_Y \widehat{P}_{\widehat{\mathcal{G}}}(Y|X) \, \mathbb{E}_{X'\sim\pi} \left[ \widehat{P}_{\widehat{\mathcal{G}}} \widehat{V}_{h+2}^\pi(X') \right] - \sum_Y P_{\mathcal{G}}(Y|X) \, \mathbb{E}_{X'\sim\pi} \left[ P_{\mathcal{G}} \widehat{V}_{h+2}^\pi(X') \right] \right| \\
&\leq \mathbb{E}_X \left| \sum_Y \left( \widehat{P}_{\widehat{\mathcal{G}}}(Y|X) - P_{\mathcal{G}}(Y|X) \right) \right| \\
&\quad + \mathbb{E}_X \left| \sum_Y \widehat{P}_{\widehat{\mathcal{G}}}(Y|X) \, \mathbb{E}_{X'\sim\pi} \left[ \widehat{P}_{\widehat{\mathcal{G}}} \widehat{V}_{h+2}^\pi(X') \right] - \sum_Y P_{\mathcal{G}}(Y|X) \, \mathbb{E}_{X'\sim\pi} \left[ P_{\mathcal{G}} \widehat{V}_{h+2}^\pi(X') \right] \right|
\end{aligned}
$$

(15)

(16)

We can now bound the first term of (16):

$$
\begin{aligned}
\mathbb{E}_X \left| \sum_Y \left( \widehat{P}_{\widehat{\mathcal{G}}}(Y|X) - P_{\mathcal{G}}(Y|X) \right) \right| &= \mathbb{E}_X \left| \sum_Y \left( \prod_{j=1}^{d_S} \widehat{P}_j(Y[j]|X[Z_j]) - \prod_{j=1}^{d_S} P_j(Y[j]|X[Z_j]) \right) \right| \\
&\leq \mathbb{E}_X \left[ \sum_Y \sum_{j=1}^{d_S} \left| \widehat{P}_j(Y[j]|X[Z_j]) - P_j(Y[j]|X[Z_j]) \right| \right] \\
&= \sum_X P_{\mathcal{G}}^\pi(X) \left[ \sum_Y \sum_{j=1}^{d_S} \left| \widehat{P}_j(Y[j]|X[Z_j]) - P_j(Y[j]|X[Z_j]) \right| \right] \\
&= \sum_Y \sum_{j=1}^{d_S} \sum_{X[Z_j]} P_{\mathcal{G}}^\pi(X[Z_j]) \left| \widehat{P}_j(Y[j]|X[Z_j]) - P_j(Y[j]|X[Z_j]) \right|
\end{aligned}
$$

(17)

Due to the uniform sampling and Z-sparseness assumptions, we have $P_{\mathcal{G}}(X[Z_j]) = \frac{1}{n^Z}$, hence:

$$
\max_{\pi^\dagger} \frac{P_{\mathcal{G}}^{\pi^\dagger}(X[Z_j])}{P_{\mathcal{G}}(X[Z_j])} \leq \frac{1}{P_{\mathcal{G}}(X[Z_j])} = n^Z
$$

Therefore:

$$
P_{\mathcal{G}}^{\pi^\dagger}(X[Z_j]) \leq n^Z \cdot P_{\mathcal{G}}(X[Z_j])
$$

Replacing this in (17) and marginalizing over $Y \setminus Y[j]$ we obtain:

$$
\begin{aligned}
\mathbb{E}_X \left| \sum_Y \left( \widehat{P}_{\widehat{\mathcal{G}}}(Y|X) - P_{\mathcal{G}}(Y|X) \right) \right| &= n^Z \sum_{j=1}^{d_s} \sum_{Y[j]} \sum_{X[Z_j]} \left| \widehat{P}_j(Y[j]|X[Z_j]) - P_j(Y[j]|X[Z_j]) \right| P_{\mathcal{G}}(X[Z_j]) \\
&\leq n^Z \sum_{j=1}^{d_S} \sum_{Y[j]} \frac{\epsilon'}{d_S} \sum_{X[Z_j]} P_{\mathcal{G}}(X[Z_j]) \\
&= n^{Z+1} \epsilon'
\end{aligned}
$$

Where $\frac{\epsilon'}{d_S}$ is the approximation term of each component. By plugging this bound into (16) we get:

$$\mathbb{E}_X \left| (\widehat{P}_{\widehat{\mathcal{G}}} - P_{\mathcal{G}}) \widehat{V}_{h+1}^\pi(X) \right| \leq n^{Z+1}\epsilon' + \mathbb{E}_X \left| \sum_Y \widehat{P}_{\widehat{\mathcal{G}}}(Y|X) \mathbb{E}_{X'\sim\pi} \left[ \widehat{P}_{\widehat{\mathcal{G}}} \widehat{V}_{h+2}^\pi(X') \right] \right.$$

$$\left. - \sum_Y P_{\mathcal{G}}(Y|X) \mathbb{E}_{X'\sim\pi} \left[ P_{\mathcal{G}} \widehat{V}_{h+2}^\pi(X') \right] \right|$$

$$\leq \sum_{i=h+1}^{H} i \cdot n^{Z+1}\epsilon'$$

$$\leq H^2 n^{Z+1}\epsilon'$$

where in the last step we have recursively bounded the right terms as in (24). By considering $2Z$-sparseness, $\lambda$-sufficiency, and that the transition model $P$ factorizes, we can apply the same procedure to bound the second term of equation (14) as:

$$\mathbb{E}_X \left| (P_{\mathcal{G}} - P) \widehat{V}_{h+1}^\pi(X) \right| \leq H^2 n^{Z+1} d_S \lambda$$

Therefore the initial expression in (13) becomes:

$$\left| \mathbb{E}_{s\sim P} \left[ \widehat{V}_1^\pi(s) - V_1^\pi(s) \right] \right| \leq \sum_{h=1}^{H} \mathbb{E}_X \left| (\widehat{P}_{\widehat{\mathcal{G}}} - P) \widehat{V}_{h+1}^\pi(X) \right| \tag{18}$$

$$\leq \sum_{h=1}^{H} [n^{Z+1}H^2\epsilon' + n^{2Z+1}d_S H^2\lambda] \tag{19}$$

$$\leq \underbrace{n^{Z+1}H^3\epsilon'}_{\epsilon} + \underbrace{n^{2Z+1}d_S H^3\lambda}_{\epsilon_\lambda} \tag{20}$$

$\square$

**Corollary B.2.** *For a tabular MDP $\mathcal{M} \in \mathbb{M}$, the result of Theorem 4.2 holds with $\epsilon_\lambda = 2\lambda SAH^3$, $\epsilon' = \epsilon/2SAH^3$.*

*Proof.* Consider the MDPs with transition model $P$ and $\widehat{P}_{\widehat{\mathcal{G}}}$. We refer to the respective optimal policies as $\pi^*$ and $\widehat{\pi}^*$. Moreover, since the reward $r$ is fixed, we remove it from the expressions for the sake of clarity, and refer with $\widehat{V}$ to the value function of the MDP with transition model $\widehat{P}_{\widehat{\mathcal{G}}}$. As done in (Jin et al., 2020, Theorem 3.5), we can write the following decomposition, where $V^* := V^{\pi^*}$.

$$\mathbb{E}_{s_1\sim P} \left[ V_1^*(s_1) - V_1^{\widehat{\pi}}(s_1) \right] \leq \underbrace{\left| \mathbb{E}_{s_1\sim P} \left[ V_1^*(s_1) - \widehat{V}_1^{\widehat{\pi}^*}(s_1) \right] \right|}_{\text{evaluation error}} + \underbrace{\mathbb{E}_{s_1\sim P} \left[ \widehat{V}_1^*(s_1) - \widehat{V}_1^{\widehat{\pi}^*}(s_1) \right]}_{\leq 0 \text{ by def.}}$$

$$+ \underbrace{\mathbb{E}_{s_1\sim P} \left[ \widehat{V}_1^{\widehat{\pi}^*}(s_1) - \widehat{V}_1^{\widehat{\pi}}(s_1) \right]}_{\text{optimization error}} + \underbrace{\left| \mathbb{E}_{s_1\sim P} \left[ \widehat{V}_1^{\widehat{\pi}}(s_1) - V_1^{\widehat{\pi}}(s_1) \right] \right|}_{\text{evaluation error}}$$

$$\leq \underbrace{2SAH^3\epsilon'}_{\epsilon} + \underbrace{2SAH^3\lambda}_{\epsilon_\lambda}$$

where in the last step we have set to 0 the approximation due to the planning oracle assumption, and we have bounded the evaluation errors according to Lemma C.4. In order to get $2SAH^3\epsilon' = \epsilon$ we have to set $\epsilon' = \frac{\epsilon}{2SAH^3}$. Considering the sample complexity result in Lemma B.1 the final sample complexity will be:

$$K = O\left( \frac{M\, S^2\, Z^2\, 2^{2Z}\, \log\left(\frac{4MS^2A2^Z}{\delta}\right)}{(\epsilon')^2} \right) = O\left( \frac{4M\, S^4\, A^2\, H^6\, Z^2\, 2^{2Z}\, \log\left(\frac{4MS^2A2^Z}{\delta}\right)}{\epsilon^2} \right)$$

$\square$

**Lemma C.4.** *Under the preconditions of Corollary B.2, with probability $1 - \delta$, for any reward function $r$ and policy $\pi$, we can bound the value function estimation error as follows.*

$$\left| \mathbb{E}_{s \sim P} \left[ \widehat{V}_{1,r}^{\pi}(s) - V_{1,r}^{\pi}(s) \right] \right| \leq \underbrace{SAH^3 \epsilon'}_{\epsilon} + \underbrace{SAH^3 \lambda}_{\epsilon_\lambda} \tag{21}$$

*where $\widehat{V}$ is the value function of the MDP with transition model $\widehat{P}_{\widehat{\mathcal{G}}}$, $\epsilon'$ is the approximation error between $\widehat{P}_{\widehat{\mathcal{G}}}$ and $P_G$ studied in Lemma 4.1, and $\lambda$ stands for the $\lambda$-sufficiency parameter of $P_{\mathcal{G}}$.*

*Proof.* The proof will be along the lines of that of Lemma 3.6 in (Jin et al., 2020). We first recall (Dann et al., 2017, Lemma E.15), which we restate in Lemma C.5. In this prof we consider an environment specific true MDP $\mathcal{M}$ with transition model $P$, and an mdp $\widehat{\mathcal{M}}$ that has as transition model the estimated causal transition model $\widehat{P}_{\widehat{\mathcal{G}}}$. In the following, the expectations will be w.r.t. $P$. Moreover, since the reward $r$ is fixed, we remove it from the expressions for the sake of clarity. We can start deriving

$$\left| \mathbb{E}_{s \sim P} \left[ \widehat{V}_1^{\pi}(s) - V_1^{\pi}(s) \right] \right| \leq \left| \mathbb{E}_{\pi} \left[ \sum_{h=1}^{H} (\widehat{P}_{\widehat{\mathcal{G}}} - P) \widehat{V}_{h+1}^{\pi}(s_h, a_h) \right] \right|$$

$$\leq \mathbb{E}_{\pi} \left[ \sum_{h=1}^{H} \left| (\widehat{P}_{\widehat{\mathcal{G}}} - P) \widehat{V}_{h+1}^{\pi}(s_h, a_h) \right| \right]$$

$$= \sum_{h=1}^{H} \mathbb{E}_{\pi} \left| (\widehat{P}_{\widehat{\mathcal{G}}} - P) \widehat{V}_{h+1}^{\pi}(s_h, a_h) \right|$$

We now bound a single term within the sum above as follows:

$$\mathbb{E}_{\pi} \left| (\widehat{P}_{\widehat{\mathcal{G}}} - P) \widehat{V}_{h+1}^{\pi}(s_h, a_h) \right| \leq \sum_{s,a} \left| (\widehat{P}_{\widehat{\mathcal{G}}} - P) \widehat{V}^{\pi}(s, a) \right| P^{\pi}(s, a)$$

$$= \sum_{s,a} \left| (\widehat{P}_{\widehat{\mathcal{G}}} - P) \widehat{V}^{\pi}(s, a) \right| P^{\pi}(s) \pi(a|s)$$

$$\leq \max_{\pi'} \sum_{s,a} \left| (\widehat{P}_{\widehat{\mathcal{G}}} - P) \widehat{V}^{\pi}(s, a) \right| P^{\pi}(s) \pi'(a|s)$$

$$= \max_{\nu: \mathcal{S} \to \mathcal{A}} \sum_{s,a} \left| (\widehat{P}_{\widehat{\mathcal{G}}} - P) \widehat{V}^{\pi}(s, a) \right| P^{\pi}(s) \mathbb{1}\{a = \nu(s)\}$$

where in the last step we have used the fact that there must exist an optimal deterministic policy. Due to the uniform sampling assumption, we have $P(s, a) = \frac{1}{SA}$, hence:

$$\max_{\pi^\dagger} \frac{P^{\pi^\dagger}(s, a)}{P(s, a)} \leq \frac{1}{P(s, a)} = SA$$

Therefore:

$$P^{\pi^\dagger}(s, a) \leq SA \cdot P(s, a)$$

Moreover, notice that, since $\pi'$ is deterministic we have $P^{\pi}(s) = P^{\pi'}(s) = P^{\pi'}(s, a) \leq SA \cdot P(s, a)$. Replacing it in the expression above we get

$$\mathbb{E}_{\pi} \left| (\widehat{P}_{\widehat{\mathcal{G}}} - P) \widehat{V}_{h+1}^{\pi}(s_h, a_h) \right| \leq SA \cdot \sum_{s,a} \left| (\widehat{P}_{\widehat{\mathcal{G}}} - P) \widehat{V}_{h+1}^{\pi}(s, a) \right| P(s) \mathbb{1}\{a = \nu(s)\}$$

$$\leq SA \cdot \left| (\widehat{P}_{\widehat{\mathcal{G}}} - P) \widehat{V}_{h+1}^{\pi}(s, a) \right|$$

$$\leq SA \cdot \left| (\widehat{P}_{\widehat{\mathcal{G}}} - P_{\mathcal{G}}) \widehat{V}_{h+1}^{\pi}(s, a) \right| + SA \cdot \left| (P_{\mathcal{G}} - P) \widehat{V}_{h+1}^{\pi}(s, a) \right| \tag{22}$$

$$\leq SA \cdot \sum_{i=h+1}^{H} i \cdot \epsilon' + SA \cdot \sum_{i=h+1}^{H} i \cdot \lambda$$

$$\leq SAH^2 \epsilon' + SAH^2 \lambda \tag{23}$$

where $\epsilon'$ is the approximation error between $\widehat{P}_{\widehat{\mathcal{G}}}$ and $P_G$ studied in Lemma 4.1, and in the penultimate step we have used the following derivation:

$$\left|(\widehat{P}_{\widehat{\mathcal{G}}} - P_{\mathcal{G}})\widehat{V}_{h+1}^{\pi}(s,a)\right| = \left|\widehat{P}_{\widehat{\mathcal{G}}}\widehat{V}_{h+1}^{\pi}(s,a) - P_{\mathcal{G}}\widehat{V}_{h+1}^{\pi}(s,a)\right| \tag{24}$$

$$= \left|\sum_{s'}\widehat{P}_{\widehat{\mathcal{G}}}(s'|s,a)\widehat{V}_{h+1}^{\pi}(s') - \sum_{s'}P_{\mathcal{G}}(s'|s,a)\widehat{V}_{h+1}^{\pi}(s')\right|$$

$$= \left|\sum_{s'}\widehat{P}_{\widehat{\mathcal{G}}}(s'|s,a)\,\mathbb{E}_{a'\sim\pi}\left[r(s',a') + \widehat{P}_{\widehat{\mathcal{G}}}\widehat{V}_{h+2}^{\pi}(s',a')\right]\right. \tag{25}$$

$$\left. - \sum_{s'}P_{\mathcal{G}}(s'|s,a)\,\mathbb{E}_{a'\sim\pi}\left[r(s',a') + P_{\mathcal{G}}\widehat{V}_{h+2}^{\pi}(s',a')\right]\right|$$

$$= \left|\sum_{s'}\left(\widehat{P}_{\widehat{\mathcal{G}}}(s'|s,a) - P_{\mathcal{G}}(s'|s,a)\right)\mathbb{E}_{a'\sim\pi}\left[r(s',a')\right]\right.$$

$$\left. + \sum_{s'}\widehat{P}_{\widehat{\mathcal{G}}}(s'|s,a)\,\mathbb{E}_{a'\sim\pi}\left[\widehat{P}_{\widehat{\mathcal{G}}}\widehat{V}_{h+2}^{\pi}(s',a')\right] - \sum_{s'}P_{\mathcal{G}}(s'|s,a)\,\mathbb{E}_{a'\sim\pi}\left[P_{\mathcal{G}}\widehat{V}_{h+2}^{\pi}(s',a')\right]\right|$$

$$\leq \epsilon' + \left|\sum_{s'}\widehat{P}_{\widehat{\mathcal{G}}}(s'|s,a)\,\mathbb{E}_{a'\sim\pi}\left[\widehat{P}_{\widehat{\mathcal{G}}}\widehat{V}_{h+2}^{\pi}(s',a')\right] - \sum_{s'}P_{\mathcal{G}}(s'|s,a)\,\mathbb{E}_{a'\sim\pi}\left[P_{\mathcal{G}}\widehat{V}_{h+2}^{\pi}(s',a')\right]\right|$$

$$= \epsilon' + \left|\sum_{s'}\widehat{P}_{\widehat{\mathcal{G}}}(s'|s,a)\,\mathbb{E}_{a'\sim\pi}\left[\sum_{s''}\widehat{P}_{\widehat{\mathcal{G}}}(s''|s',a')\,\mathbb{E}_{a''\sim\pi}\left[r(s'',a'') + \widehat{P}_{\widehat{\mathcal{G}}}\widehat{V}_{h+3}^{\pi}(s'',a'')\right]\right]\right.$$

$$\left. - \sum_{s'}P_{\mathcal{G}}(s'|s,a)\,\mathbb{E}_{a'\sim\pi}\left[\sum_{s''}P_{\mathcal{G}}(s''|s',a')\,\mathbb{E}_{a''\sim\pi}\left[r(s'',a'') + P_{\mathcal{G}}\widehat{V}_{h+3}^{\pi}(s'',a'')\right]\right]\right|$$

$$\leq \epsilon' + \sum_{s',s'',a'}\left|\widehat{P}_{\widehat{\mathcal{G}}}(s'|s,a)\widehat{P}_{\widehat{\mathcal{G}}}(s''|s',a') - P_{\mathcal{G}}(s'|s,a)P_{\mathcal{G}}(s''|s',a')\right|_1 + \dots$$

$$\leq \epsilon' + \sum_{s',s'',a'}\left[\left|\widehat{P}_{\widehat{\mathcal{G}}}(s'|s,a) - P_{\mathcal{G}}(s'|s,a)\right|_1 + \left|\widehat{P}_{\widehat{\mathcal{G}}}(s''|s',a') - P_{\mathcal{G}}(s''|s',a')\right|_1\right] + \dots$$

$$\leq \epsilon' + 2\epsilon' + \dots$$

Hence, due to this recursive unrolling, we have:

$$\left|(\widehat{P}_{\widehat{\mathcal{G}}} - P_{\mathcal{G}})\widehat{V}_{h+1}^{\pi}(s,a)\right| \leq \sum_{i=h+1}^{H} i\epsilon' \leq H^2\epsilon$$

Notice that the same argument holds also for the second term of (22), replacing $\epsilon'$ with $\lambda$.

By plugging the result in equation (23) into the initial expression we get:

$$\left|\mathbb{E}_{s\sim P}\left[\widehat{V}_1^{\pi}(s) - V_1^{\pi}(s)\right]\right| \leq \sum_{h=1}^{H}\mathbb{E}_{\pi}\left|(\widehat{P}_{\widehat{\mathcal{G}}} - P)\widehat{V}_{h+1}^{\pi}(s_h, a_h)\right|$$

$$\leq \sum_{h=1}^{H} SAH^2\epsilon' + SAH^2\lambda$$

$$= SAH^3\epsilon' + SAH^3\lambda$$

$\square$

In the following we restate (Dann et al., 2017, Lemma E.15) for the case of stationary transition model.

**Lemma C.5.** *For any two MDPs $\mathcal{M}'$ and $\mathcal{M}''$ with rewards $r'$ and $r''$ and transition models $P'$ and $P''$, the difference in value functions $V', V''$ w.r.t. the same policy $\pi$ can be written as:*

$$V_h'(s) - V_h''(s) = \mathbb{E}_{\mathcal{M}'',\pi}\left[\sum_{i=h}^{H}[r'(s_i, a_i) - r''(s_i, a_i) + (P' - P'')V_{i+1}'(s_i, a_i)] \mid s_h = s\right] \tag{26}$$

PROOFS OF SECTION B.1

We provide the proof of the sample complexity result for learning the causal structure of a discrete MDP with a generative model.

**Theorem B.4.** *Let $\mathcal{M}$ be a discrete MDP with an underlying causal structure $\mathcal{G}$, let $\delta \in (0, 1)$, and let $\epsilon > 0$. The Algorithm 2 returns a dependency graph $\widehat{\mathcal{G}}$ such that $Pr(\widehat{\mathcal{G}} \neq \mathcal{G}_\epsilon) \leq \delta$ with a sample complexity*

$$K = O\big(n \log(d_S^2 d_A / \delta) / \epsilon^2\big).$$

*Proof.* We aim to obtain the number of samples $K$ for which Algorithm 2 is guaranteed to return a causal structure estimate $\widehat{\mathcal{G}}$ such that $Pr(\widehat{\mathcal{G}} \neq \mathcal{G}_\epsilon) \leq \delta$ in a discrete MDP setting. First, we can upper bound the probability of the bad event $Pr(\widehat{\mathcal{G}} \neq \mathcal{G}_\epsilon)$ in terms of the probability of a failure in the independence testing procedure $\mathbb{I}(X_z, Y_j)$ for a single pair of nodes $X_z \in \mathcal{G}_\epsilon, Y_z \in \mathcal{G}_\epsilon$, i.e.,

$$Pr(\widehat{\mathcal{G}} \neq \mathcal{G}_\epsilon) \leq Pr\bigg( \bigcup_{z=1}^{d_S + d_A} \bigcup_{j=1}^{d_S} \text{ test } \mathbb{I}(X_z, Y_j) \text{ fails} \bigg) \leq \sum_{z=1}^{d_S + d_A} \sum_{j=1}^{d_S} Pr\bigg( \text{test } \mathbb{I}(X_z, Y_j) \text{ fails} \bigg), \tag{27}$$

where we applied an union bound to obtain the last inequality. Now we can look at the probability of a single independence test failure. Especially, for a provably efficient independence test (the existence of such a test is stated by Lemma B.3, whereas the Algorithm 2 in Diakonikolas et al. (2021) reports an actual testing procedure), we have $Pr(\text{test } \mathbb{I}(X_z, Y_j) \text{ fails}) \leq \delta'$, for any choice of $\delta' \in (0, 1)$, $\epsilon' > 0$, with a number of samples

$$K' = C\bigg( \frac{n \, \log^{1/3}(1/\delta')}{(\epsilon')^{4/3}} + \frac{n \, \log^{1/2}(1/\delta') + \log(1/\delta')}{(\epsilon')^2} \bigg), \tag{28}$$

where $C$ is a sufficiently large universal constant (Diakonikolas et al., 2021, Theorem 1.3). Finally, by letting $\epsilon' = \epsilon$, $\delta' = \frac{\delta}{d_S^2 d_A}$ and combining (27) with (28), we obtain $Pr(\widehat{\mathcal{G}} \neq \mathcal{G}_\epsilon)$ with a sample complexity

$$K = O\bigg( \frac{n \log(d_S^2 d_A / \delta)}{\epsilon^2} \bigg),$$

under the assumption $\epsilon^2 \ll \epsilon^{4/3}$, which concludes the proof. $\square$

The proof of the analogous sample complexity result for a tabular MDP setting (Corollary B.5) is a direct consequence of Theorem B.4 by letting $n = 2, d_S = S, d_A = A$.

PROOFS OF SECTION B.2

We first report a useful concentration inequality for the L1-norm between the empirical distribution computed over $K$ samples and the true distribution (Weissman et al., 2003, Theorem 2.1).

**Lemma C.6** (Weissman et al. (2003)). *Let $X_1, \ldots, X_K$ be i.i.d. random variables over $[n]$ having probabilities $Pr(X_k = i) = P_i$, and let $\widehat{P}_K(i) = \frac{1}{K} \sum_{k=1}^{K} \mathbb{1}(X_k = i)$. Then, for every threshold $\epsilon > 0$, it holds*

$$Pr\bigg( \|\widehat{P}_K - P\|_1 \geq \epsilon \bigg) \leq 2 \exp(-K\epsilon^2 / 2n).$$

We can now provide the proof of the sample complexity result for learning the Bayesian network of a discrete MDP with a given causal structure.

**Theorem B.6.** *Let $\mathcal{M}$ be a discrete MDP, let $\mathcal{G}$ be its underlying causal structure, let $\delta \in (0, 1)$, and let $\epsilon > 0$. The Algorithm 3 returns a Bayesian network $\widehat{P}_\mathcal{G}$ such that $Pr(\|\widehat{P}_\mathcal{G} - P_\mathcal{G}\|_1 \geq \epsilon) \leq \delta$ with a sample complexity*

$$K = O\big(d_S^3 n^{3Z+1} \log(d_S n^Z / \delta) / \epsilon^2\big).$$

*Proof.* We aim to obtain the number of samples $K$ for which Algorithm 3 is guaranteed to return a Bayesian network estimate $\widehat{P}_{\mathcal{G}}$ such that $Pr(\|\widehat{P}_{\mathcal{G}} - P_{\mathcal{G}}\|_1 \geq \epsilon) \leq \delta$ in a discrete MDP setting. First, we note that

$$Pr\Big(\|\widehat{P}_{\mathcal{G}} - P_{\mathcal{G}}\|_1 \geq \epsilon\Big) \leq Pr\Big(\sum_{j=1}^{d_S} \|\widehat{P}_j - P_j\|_1 \geq \epsilon\Big) \tag{29}$$

$$\leq Pr\Big(\frac{1}{d_S}\sum_{j=1}^{d_S} \|\widehat{P}_j - P_j\|_1 \geq \frac{\epsilon}{d_S}\Big) \tag{30}$$

$$\leq Pr\Big(\bigcup_{j=1}^{d_S} \|\widehat{P}_j - P_j\|_1 \geq \frac{\epsilon}{d_S}\Big) \tag{31}$$

$$\leq \sum_{j=1}^{d_S} Pr\Big(\|\widehat{P}_j - P_j\|_1 \geq \frac{\epsilon}{d_S}\Big), \tag{32}$$

in which we employed the property $\|\mu - \nu\|_1 \leq \|\prod_i \mu_i - \prod_i \nu_i\|_1 \leq \sum_i \|\mu_i - \nu_i\|_1$ for the L1-norm between product distributions $\mu = \prod_i \mu_i, \nu = \prod_i \nu_i$ to write (29), and we applied a union bound to derive (32) from (31). Similarly, we can write

$$Pr\Big(\|\widehat{P}_j - P_j\|_1 \geq \frac{\epsilon}{d_S}\Big) \leq Pr\Big(\bigcup_{x \in [n]^{|Z_j|}} \|\widehat{P}_j(\cdot|x) - P_j(\cdot|x)\|_1 \geq \frac{\epsilon}{d_S n^{|Z_j|}}\Big) \tag{33}$$

$$\leq \sum_{x \in [n]^{|Z_j|}} Pr\Big(\|\widehat{P}_j(\cdot|x) - P_j(\cdot|x)\|_1 \geq \frac{\epsilon}{d_S n^{|Z_j|}}\Big) \tag{34}$$

$$\leq \sum_{x \in [n]^{|Z_j|}} Pr\Big(\|\widehat{P}_j(\cdot|x) - P_j(\cdot|x)\|_1 \geq \frac{\epsilon}{d_S n^{Z}}\Big) \tag{35}$$

by applying a union bound to derive (34) from (33), and by employing Assumption 1 to bound $|Z_j|$ with $Z$ in (35). We can now invoke Lemma C.6 to obtain the sample complexity $K'$ that guarantees $Pr(\|\widehat{P}_j(\cdot|x) - P_j(\cdot|x)\|_1 \geq \epsilon') \leq \delta'$, i.e.,

$$K' = \frac{2n\log(2/\delta')}{(\epsilon')^2} = \frac{2\,d_S^2\,n^{2Z+1}\,\log(2d_S n^Z/\delta)}{\epsilon^2},$$

where we let $\epsilon' = \frac{\epsilon}{d_S n^Z}$, $\delta' = \frac{\delta}{d_S n^Z}$. Finally, by summing $K'$ for any $x \in [nm]^{|Z_j|}$ and any $j \in [d_S]$, we obtain

$$K = \sum_{j \in [d_S]} \sum_{x \in [n]^{|Z_j|}} K' \leq \frac{2\,d_S^3\,n^{3Z+1}\,\log(2d_S n^Z/\delta)}{\epsilon^2},$$

which proves the theorem. $\qquad\square$

To prove the analogous sample complexity result for a tabular MDP we can exploit a slightly tighter concentration on the KL divergence between the empirical distribution and the true distribution in the case of binary variables (Dembo & Zeitouni, 2009, Theorem 2.2.3)[6], which we report for convenience in the following lemma.

**Lemma C.7** (Dembo & Zeitouni (2009)). *Let $X_1, \ldots, X_K$ be i.i.d. random variables over $[2]$ having probabilities $Pr(X_k = i) = P_i$, and let $\widehat{P}_K(i) = \frac{1}{K}\sum_{k=1}^K \mathbb{1}(X_k = i)$. Then, for every threshold $\epsilon > 0$, it holds*

$$Pr\Big(d_{KL}\big(\widehat{P}_K\|P\big) \geq \epsilon\Big) \leq 2\exp(-K\epsilon).$$

We can now provide the proof of Corollary B.7.

---

[6]Also reported in (Mardia et al., 2020, Example 1).

**Corollary B.7.** *Let $\mathcal{M}$ be a tabular MDP. The result of Theorem B.6 reduces to $K = O\big(S^2 2^{2Z} \log(S2^Z/\delta)/\epsilon^2\big)$.*

*Proof.* We aim to obtain the number of samples $K$ for which Algorithm 3 is guaranteed to return a Bayesian network estimate $\widehat{P}_{\mathcal{G}}$ such that $Pr(\|\widehat{P}_{\mathcal{G}} - P_{\mathcal{G}}\|_1 \geq \epsilon) \leq \delta$ in a tabular MDP setting. We start by considering the KL divergence $d_{KL}\big(\widehat{P}_{\mathcal{G}}\|P_{\mathcal{G}}\big)$. Especially, we note

$$
\begin{aligned}
d_{KL}\big(\widehat{P}_{\mathcal{G}}\|P_{\mathcal{G}}\big) &= \sum_{X,Y} \widehat{P}_{\mathcal{G}}(X,Y) \log \frac{\widehat{P}_{\mathcal{G}}(X,Y)}{P_{\mathcal{G}}(X,Y)} \\
&= \sum_{X,Y} \widehat{P}_{\mathcal{G}}(X,Y) \log \frac{\prod_{j=1}^{S} \widehat{P}_j(Y[j]|X[Z_j])}{\prod_{j=1}^{S} P_j(Y[j]|X[Z_j])} \\
&= \sum_{X,Y} \widehat{P}_{\mathcal{G}}(X,Y) \sum_{j=1}^{S} \log \frac{\widehat{P}_j(Y[j]|X[Z_j])}{P_j(Y[j]|X[Z_j])} = \sum_{j=1}^{S} d_{KL}\big(\widehat{P}_j\|P_j\big).
\end{aligned}
$$

Then, for any $\epsilon' > 0$ we can write

$$
Pr\Big(d_{KL}\big(\widehat{P}_{\mathcal{G}}\|P_{\mathcal{G}}\big) \geq \epsilon'\Big) \leq Pr\bigg(\bigcup_{j=1}^{S} d_{KL}\big(\widehat{P}_j\|P_j\big) \geq \frac{\epsilon'}{S}\bigg) \tag{36}
$$

$$
\leq \sum_{j=1}^{S} Pr\bigg(d_{KL}\big(\widehat{P}_j\|P_j\big) \geq \frac{\epsilon'}{S}\bigg) \tag{37}
$$

$$
\leq \sum_{j=1}^{S} Pr\bigg(\bigcup_{x \in [2]^{|Z_j|}} d_{KL}\big(\widehat{P}_j(\cdot|x)\|P_j(\cdot|x)\big) \geq \frac{\epsilon'}{S2^{|Z_j|}}\bigg) \tag{38}
$$

$$
\leq \sum_{j=1}^{S} \sum_{x \in [2]^{|Z_j|}} Pr\bigg(d_{KL}\big(\widehat{P}_j(\cdot|x)\|P_j(\cdot|x)\big) \geq \frac{\epsilon'}{S2^{|Z_j|}}\bigg) \tag{39}
$$

$$
\leq \sum_{j=1}^{S} \sum_{x \in [2]^{|Z_j|}} Pr\bigg(d_{KL}\big(\widehat{P}_j(\cdot|x)\|P_j(\cdot|x)\big) \geq \frac{\epsilon'}{S2^{Z}}\bigg), \tag{40}
$$

in which we applied a first union bound to get (37) from (36), a second union bound to get (39) from (38), and Assumption 1 to bound $|Z_j|$ with $Z$ in (40). We can now invoke Lemma C.7 to obtain the sample complexity $K''$ that guarantees $Pr(d_{KL}(\widehat{P}_j(\cdot|x)\|P_j(\cdot|x)) \geq \epsilon'') \leq \delta''$, i.e.,

$$
K'' = \frac{\log(2/\delta'')}{\epsilon''} = \frac{S2^Z \log(2S2^Z/\delta')}{\epsilon'},
$$

where we let $\epsilon'' = \frac{\epsilon'}{S2^Z}$, and $\delta'' = \frac{\delta'}{S2^Z}$ for any choice of $\delta' \in (0,1)$. By summing $K''$ for any $x \in [2]^{|Z_j|}$ and and $j \in [S]$, we obtain the sample complexity $K'$ that guarantees $Pr(d_{KL}(\widehat{P}_{\mathcal{G}}\|P_{\mathcal{G}}) \geq \epsilon') \leq \delta'$, i.e.,

$$
K' = \sum_{j=1}^{S} \sum_{x \in [2]^{|Z_j|}} K'' \leq \frac{S^2 2^{2Z} \log(2S2^Z/\delta')}{\epsilon'}. \tag{41}
$$

Finally, we employ the Pinsker's inequality $\|\widehat{P}_{\mathcal{G}} - P_{\mathcal{G}}\|_1 \leq \sqrt{2d_{KL}(\widehat{P}_{\mathcal{G}}\|P_{\mathcal{G}})}$ Csiszár (1967) to write

$$
Pr\Big(d_{KL}\big(\widehat{P}_{\mathcal{G}}\|P_{\mathcal{G}}\big) \geq \epsilon'\Big) = Pr\Big(\sqrt{2d_{KL}(\widehat{P}_{\mathcal{G}}\|P_{\mathcal{G}})} \geq \sqrt{2\epsilon'}\Big) \geq Pr\Big(\|\widehat{P}_{\mathcal{G}} - P_{\mathcal{G}}\|_1 \geq \sqrt{2\epsilon'}\Big),
$$

which gives the sample complexity $K$ that guarantees $Pr(\|\widehat{P}_{\mathcal{G}} - P_{\mathcal{G}}\|_1 \geq \epsilon) \leq \delta$ by letting $\epsilon' = \frac{\epsilon^2}{2}$ and $\delta' = \delta$ in (41), i.e.,

$$
K = \frac{2S^2 2^{2Z} \log(2S2^Z/\delta)}{\epsilon^2}.
$$

$\square$

