# OpenReview forum: "Provably Efficient Causal Model-Based Reinforcement Learning for Systematic Generalization"
_ICLR.cc/2022/Workshop/OSC — Submitted to ICLR2022 OSC _

### Official Review · Reviewer_z9a7 · 2022-03-15
**Possibly interesting and relevant theoretical paper --- unfortunately I lack the expertise to assess this**

**Rating:** 2
**Confidence:** 1

**Review:**

### Summary

The focus of this work is on systematic generalization in RL, which is here used to refer to generalizing universal causal relations (such as laws of physics) learned from interacting with a few environment to approximately solve any task in any other environment without further interactions. This is made tractable through a set of structural assumptions about the shared causal structure that explains a significant portion of the transition dynamics in these reward-free environments. When using a causal transition model (obtained by estimating a Bayesian network from a mixture of the environments) to infer the common dynamics, then it can be employed by a planning oracle to provide an approximately optimal policy for a latent environment and a given reward function.

Here it is shown that this simple recipe allows achieving any desired planning error up to an unavoidable term inherent to the setting. An analysis of the sample complexity of this approach is provided, which is polynomial in all relevant quantities of the problem.

### Review

Unfortunately, I am not an expert in RL theory, which makes it difficult for me to evaluate the main contribution, which is Theorem 4.2, and consequently any of the claims above. Given my unfamiliarity, it seems unreasonable to expect me to review the lengthy derivation in the appendix for a workshop. I apologize to the authors for my lack of feedback, insights, or comments, especially since the work appears both relevant and interesting.

---

### Official Review · Reviewer_xSZu · 2022-03-16

**Rating:** 1
**Confidence:** 1

**Review:**

Overall, it was difficult to understand the main approach presented in the paper.
It was unclear as to the kind of environments that were being considered in the approach. The presentation could have benefitted significantly by considering a few standard environments / domains; then following that with the approach and discussing each component of the approach.

---

### Decision · Program_Chairs · 2022-03-24

**Decision:**

Reject

**Comment:**

As highlighted by the reviewers, this seem like a relevant and interesting contribution. As highlighted by reviewer xSZuVivek, this work could benefit from some more clarity in presentation. While the authors do a little to motivate learning about the causal dynamics of the world in the introduction, this work would benefit from relating the assumptions and results to standard environments/domains. We recommend submitting this work to a later venue.